# Very Early Response Evaluation by PET/MR in Patients with Lung Cancer—Timing and Feasibility

**DOI:** 10.3390/diagnostics9010035

**Published:** 2019-03-26

**Authors:** Natasha Hemicke Langer, Seppo W. Langer, Helle Hjorth Johannesen, Adam Espe Hansen, Junia Costa, Thomas Levin Klausen, Julie Forman, Anders Olin, Sine Hvid Rasmussen, Jens Benn Sørensen, Johan Löfgren, Andreas Kjær, Barbara Malene Fischer

**Affiliations:** 1Dept. of Clinical Physiology, Nuclear Medicine & PET and Cluster for Molecular Imaging, Rigshospitalet, Copenhagen University Hospital, Blegdamsvej 9, 2100 Copenhagen, Denmark; Natasha@langer.dk (N.H.L.); hellehjorth@dadlnet.dk (H.H.J.); adam.espe.hansen@regionh.dk (A.E.H.); thomas.levin.klausen@regionh.dk (T.L.K.); anders.olin@regionh.dk (A.O.); sinehvid@gmail.com (S.H.R.); Johan.Loefgren@regionh.dk (J.L.); Andreas.Kjaer@regionh.dk (A.K.); 2Dept. of Oncology, Rigshospitalet, Copenhagen University Hospital, Blegdamsvej 9, 2100 Copenhagen, Denmark; seppolanger@dadlnet.dk (S.W.L.); jens.benn.soerensen@regionh.dk (J.B.S.); 3Dept. of Radiology, Rigshospitalet, Blegdamsvej 9, 2100 Copenhagen, Denmark; Junia.Cardoso.Costa@regionh.dk; 4Section of Biostatistics, Dept. of Public Health, Faculty of Health Sciences, University of Copenhagen, 2200 Copenhagen, Denmark; jufo@sund.ku.dk; 5PET Centre, School of Biomedical Engineering and Imaging Sciences, Kings College Hospital, Guy’s & St Thomas Hospital, SE1 9RT London, UK

**Keywords:** response evaluation, lung cancer, non-small-cell lung carcinoma (NSCLC), FDG-PET, diffusion weighted magnetic resonance imaging (DW-MRI)

## Abstract

**Purpose:** With the increasing number of therapy options available for patients with lung cancer, early response evaluation is needed. We performed this pilot study to assess the feasibility of early, repeated Positron emission tomography-magnetic resonance (PET/MR), the impact of timing and the capability for response prediction in lung tumors during chemotherapy. **Methods:** Patients with stage IV non-small cell lung cancer referred for chemotherapy were prospectively recruited. Fluorine-18-Fluorodeoxyglucose(^18^F-FDG)-PET/MR scans were performed prior to, during and after the first or second cycle of chemotherapy. Primary tumors were defined on all scans and size, FDG-uptake and apparent diffusion coefficient (ADC) were measured. Early response was described over time and a Standard Linear Mixed Model was applied to analyze changes over time. **Results:** 45 FDG-PET/MR scans were performed in 11 patients. Whereas the overall changes measured by ADC did not change significantly, there was an overall significant decrease in FDG-uptake from pre to post treatment scans. There was no difference in the FDG-uptake measured 1 or 3 weeks after therapy, but uptake measured 2 weeks after therapy differed from measurements at week 3. Changes measured in patients scanned during the first treatment cycle appeared more pronounced than during the second cycle. **Conclusions:** This pilot study indicates that response evaluation shortly after initiation of chemotherapy appears concordant with later evaluation and probably more reliable than evaluation midway between cycles. Responses during or after the first cycle of chemotherapy rather than during subsequent cycles are likely to be more readily measured.

## 1. Introduction

Despite increased focus on early detection, most patients with non-small cell lung cancer (NSCLC) present with advanced disease, i.e., they are inoperable and a cure is extremely rare [1]. If not eligible for first-line targeted therapy or immunotherapy, these patients may be offered palliative chemotherapy with cisplatin or carboplatin combined with a third-generation drug, e.g., pemetrexed, paclitaxel, or vinorelbine. Response during chemotherapy is evaluated by tumor shrinkage on computed tomography (CT) after two or three cycles of chemotherapy in agreement with the response evaluation criteria in solid tumors (RECIST) [2]. As each treatment cycle usually lasts for three weeks, the first response evaluation is typically carried out eight to twelve weeks after the initiation of cytotoxic treatment. The median survival time for patients with advanced NSCLC is 3–4 months without treatment and about 10–11 months when receiving palliative chemotherapy [1]. Further improvement in survival for this group of patients will require new treatment strategies [3] and more efficient follow-up to make the right decisions about which patients would benefit from chemotherapy and who are unlikely to benefit and should be spared the side effects. Early evaluation of treatment response enabling modification of ineffective treatment regimens is therefore of great interest. 

The combination of positron emission tomography and computed tomography with ^18^F-fluorodeoxyglucose (FDG-PET/CT) has been a game changer in diagnosis, staging, and treatment planning in cancer patients since its introduction in 2001 [4,5] and is a well-documented and widely used imaging modality for the diagnosis and staging of patients with lung cancer [6,7]. However, despite numerous studies on response assessment indicating a possible role for FDG-PET/CT, it is rarely used for response evaluation in lung cancer. The true value of FDG-PET for monitoring treatment response is impeded by the diversity and heterogeneity of the published data [8], including challenges associated with standardization of fasting, timing between injection and scan as well as differences in changes in FDG uptake depending on the type of therapy [9,10,11]. Further, some studies have discouraged the use of FDG-PET earlier than 2 weeks after last chemotherapy due to the risk of flare [12]. Few studies have actually investigated the correlation between FDG uptake, and possible response and timing of PET after therapy [13]. The recommendations are based on in vitro studies and studies in mice with implanted tumors, suggesting the existence of (1) a flare-phenomenon [14,15,16,17] where an increased uptake of FDG in the cells is seen as a short-term effect of the treatment without reflecting lack of efficacy, and (2) stunning, where the retention of FDG in malignant cells temporarily decreases without a reflecting response [18]. 

No conclusive data on the optimal interval between chemotherapy and FDG-PET scan have yet been published. Nonetheless, these early and scarce data still have a huge impact on the possible use of FDG-PET for response evaluation in solid tumors, discouraging very early response evaluation and obstructing smooth logistics.

The combination of PET and magnetic resonance imaging (MRI), PET/MRI, was introduced in 2010 as an integrated clinical imaging modality [19,20,21]. Compared to PET/CT, the PET/MRI system enables radiation dose reduction, improved soft-tissue contrast and importantly simultaneously acquisition of information on tumor anatomy and several functional parameters [19,21,22]. However, without a CT transmission scan, a major technical challenge for PET/MRI is attenuation correction. Methods, implemented clinically, to overcome this result in incorrect PET quantification [23]. In this respect, diffusion-weighted MR imaging (DW-MRI) is a widely used non-invasive, quantitative technique that, similar to FDG-PET, has shown promise as a tool in early response evaluation of cytotoxic treatment [24]. DW-MRI has gained increasing use in staging and therapy evaluation of cancer [25,26]. DW-MRI reflects the cell density and is based on diffusion of water molecules in tissues [27]. DW-MRI can be quantified by calculation of ADC (apparent diffusion coefficient). The more undamaged cells in a tissue, the higher the restriction of water diffusion and the lower the value of ADC.

The overriding aim of this study was to explore if very early (i.e., within the first week of treatment) response evaluation using multi-parametric imaging is possible. As a first step, this pilot study was performed with the following three-fold purpose: (A) to test the feasibility of repeated PET/MR early during chemotherapy, (B) to examine whether the information on tumor characteristic simultaneously acquired on FDG-PET and DW-MRI reveals patterns that potentially enable us to predict responses very early during therapy and (C) to explore the postulated existence of a flare-phenomenon in malignant lung tumors early during chemotherapy, which could potentially exclude early response evaluation.

## 2. Methods

### 2.1. Study Population 

Patients with advanced NSCLC referred to the Department of Oncology at Rigshospitalet (Copenhagen, Denmark) for standard first line palliative chemotherapy with carboplatin and vinorelbine were included. Patients in performance status 0–1, with no known contraindications to MRI, could enter the study before the first or the second cycle of chemotherapy. 

Patients were asked to undergo up to five FDG-PET/MR scans during one cycle of chemotherapy: Scan 1: 0–3 days prior to initiation of chemotherapy, Scan 2: day 1–3 after chemotherapy, Scan 3: day 6 or 7 (prior to administration of oral vinorelbine on day 8), Scan 4: day 8–11, and Scan 5: day 18–21 (1–2 days prior to initiating the next cycle). The study was approved by the departmental science committees at Rigshospitalet, by the Regional Ethics Committee, approval number H-3-2013-090, and by the Danish Data Protection Agency. 

### 2.2. Imaging

All patients were scanned on an integrated PET/MR system (Siemens Biograph mMR, Erlangen, Germany) with a 3 Tesla magnet using a combination of spine and flexible body coils. Patients were instructed to fast for minimum six hours prior to attendance. They were scanned 60 min after injection of FDG (2 MBq/kg). 

PET was performed over 2 bed-positions covering the chest and upper abdomen, each bed-position with 8 min acquisition time. MRI included the following sequences: T1 VIBE transaxial, T2 HASTE transaxial and T2 BLADE coronal (all in breath hold) and Dixon for MR-based attenuation correction (MR-AC) of PET data. The Dixon sequence was repeated twice without repositioning, resulting in two (A and B) MR-AC maps [28]. DW-MRI was acquired using a single-shot EPI with b-values 150 and 1000, TR = 10300 ms, TE = 73 ms, parallel imaging factor = 2, voxel size 3.0 × 3.0 mm^2^, slice thickness/gap 5/1 mm, 34 slices/bed. EPI distortions were corrected using FSL (www.fmrib.ox.ac.uk/fsl) with FUGUE algorithm. The pre-therapy scan was performed after injection of iv gadolinium contrast agent (0.1 mL/kg, Gadovist 1 mmol/mL, Bayer, Berlin, Germany) before acquisition of a T1 VIBE fat saturated transaxial MRI sequence. 

PET data was reconstructed using 3-dimensional ordinary Poisson ordered-subset expectation maximization with 3 iterations, 21 subsets, and 4-mm Gaussian post filtering on 344 · 344 · 224 matrices with a voxel size of 2.1 · 2.1 · 2.0 mm.

### 2.3. PET/MRI Reading and Data Extraction

Two experienced PET/MR physicians reviewed all MR-AC maps. In general, MR-AC map A was used for attenuation correction of the PET images. MR-AC map B was used for attenuation correction only if MR-AC map A was subjected to artifacts, and MR-AC maps B was not. Differences between the MR-AC maps were mainly caused by as respiration motion and patient movement during scan (described in details by Olin et al. [28]). The tumor volumes (primary T-site) were defined and segmented on DW-MRI and PET images by an experienced radiologist respectively nuclear medicine physician. This was done using image analysis tool Mirada XD3 (Mirada Medical, Oxford, UK) and the following parameters were extracted: SUV_max,_ SUV_mean_ (maximum and mean standardized uptake value normalized to injected dose and body weight), PERCIST measures [29] including SUL_peak_ (peak standardized uptake value normalized to injected dose and lean body mass), liver reference uptake and standard deviation as well as tumor size (longest diameter on axial T2 images), and ADC_mean_ and ADC_median_ (mean and median ADC acquired from ADC maps based on b150–b1000). 

Standardized uptake values were calculated as follows:(1)$$\mathrm{SUV}=\frac{r}{\left(\frac{\;a\;\prime }{w}\right)}$$

For calculating SUV_max_, r is the maximum radioactivity activity concentration [kBq/mL] measured by the PET scanner in any voxel within the tumor. For SUVmean r is the mean radioactivity activity concentration [kBq/mL] measured in the tumor. a′ is the decay-corrected amount of injected radiolabeled FDG [kBq], and w is the weight of the patient in grams.

For SULpeak, r is the radioactivity concentration averaged with a 10-mm-diameter spherical ROI positioned by the MIRADA software (https://mirada-medical.com) within the tumor so as to maximize the enclosed average. Instead of normalizing to the weight of the patient, the lean body mass (LBM) of the patient was inserted in the above formula. 

### 2.4. Statistical Analysis

The study was designed as a feasibility study. In order to enable an examination of trends in SUV and ADC parameters over several time points in a Linear Mixed Model, the inclusion of 10 patients was estimated as the minimum requirement. 

Descriptive statistics were reported as mean ± SD for normally distributed data and median with inter quartile range for skewed data. A visual inspection of histograms was used to assess normal distribution of data. If normal distribution was not met, data was transformed using the natural logarithm prior to analysis. Possible correlations between variables were evaluated using Pearson’s correlation coefficient. A Standard Linear Mixed Model (LMM) with scan as fixed effect and subject as random effect was applied, analyzing potential changes in outcome over time. The LMM implicitly imputes missing values and provides optimal unbiased inference for longitudinal data under the assumption that missing data is *missing at random*. To examine possible response patterns, the datasets were stratified into response categories according to RECIST 1.1 criteria based on CT evaluation after 2–3 cycles of therapy, and, into 1st cycle patients and 2nd cycle patients. For PET the PERCIST criteria was applied in which a decrease in SUL_peak_ equal to or greater than 30% (and at least 0.8 SUL unit) equals partial metabolic response, a similar increase of at least 30% equals progressive metabolic disease and changes in SUL_peak_ within ± 30% indicates stable metabolic disease [29]. Similar recommendations are not available for DW-MRI, but based on the recent work by Weller et al. [30] we chose an increase in ADC values equal to or greater than 25% as indicative of response, values between ± 25% as no change, and a decrease in ADC greater or equal to 25% as progression. Data handling and statistically analyses were performed using SPSS (IBM corporation, Armonk, NY, USA, version 6.3), the LMM analyses were performed with SAS statistical software (SAS Enterprise guide version 6.1, SAS Institute Inc, Cary, NC, USA). P-values below 0.05 was valued significant. However, since this study was designed as a feasibility study, differences attaining a higher level of significance (*p* < 0.1) are also reported and no correction for multiple comparisons was made. 

## 3. Results

### 3.1. Patients

Successive patients referred from October 2013 to September 2015 for treatment of stage IV NSCLC were pre-screened. After screening at first visit, 81 patients were found eligible. Informed consent was obtained from 29, whereas 52 declined. Among the 29 patients, 16 patients were cancelled before or during the first scan due to logistical or technical problems. Two patients got anxious in the scanner and were excluded. Thus, eleven patients with stage IV NSCLC completed at least two scans. Patient characteristics are summarized in Table 1. All patients were followed up with an evaluating CT as part of the standard program after two or three cycles of chemotherapy in agreement with the response evaluation criteria in solid tumors (RECIST 1.1). All patients were followed until death or for at least 24 months. 

### 3.2. PET/MR-Scans

A total of 45 PET/MRI scans were performed. Six patients were scanned during the first cycle of chemotherapy, five during the second cycle. Table 2 summarizes the timing of the scans among patients, as well as the percentage of attendance. All PET/MR scans were completed successfully except DW-MRI on day 2 in patient number 1. Artifacts were seen in the MR-AC map A but not in map B in 4 patients. In these cases, the MR-AC map B was used for attenuation correction. Tumor size measured on MR decreased slightly but significantly from scan 1 (day 1) to scan 5 (day 19). There was no significant correlation between any SUV and ADC measures. However, ADC_mean_ and ADC_median_ were strongly correlated (*ρ* = 0.987, *p* < 0.001). Similarly, the three SUV measures (SUV_mean_, SUV_max_ and SUL_peak_) correlated strongly (*ρ* ≥ 0.9, *p* < 0.001). Thus, in the following only results for ADC_median_, SUV_max_ and SUL_peak_ will be presented. ADCmedian was chosen over ADCmean, as this value is less sensitive to noise and e.g pixels with ADC = 0 (due to missing signal). SUV_max_ was chosen as this is the most frequently reported SUV measure in the literature, also it is known to be robust with regard to segmentation and interobserver variability. However, since the size of a single voxel varies considerably among PET systems and results in various noise levels in the metric, SUV_max_ is subject to bias favoring the use of SUV_peak_. SUL_peak_ is also the recommend measure for quantification of changes in FDG-uptake according to PERCIST (PET response criteria in solid tumors) [29].

### 3.3. Changes in SUV

Illustration of absolute SUV-values and relative changes for individual patients during the course of therapy are available in Appendix A. Inspection of histograms (not shown) revealed that SUV-data were almost normally distributed. Deviation from normality was due to a few outliers and did not improve after transformation. 

Figure 1A illustrates changes in FDG-uptake during the study period for individual patients. Table 3 illustrates mean and standard deviation for SUV values at each time point for all patients and grouped by response respectively whether patients were scanned during first or second cycle of chemotherapy. Due to the low number of patients in each group no statistical comparison between groups was performed, but results suggest higher SUV-values and more pronounced changes in patients scanned during the first treatment cycle as compared to second cycle.

The LMM analysis revealed a small but significant fall (all patients) in both SUV_max_ (−1.8, 95% Confidence interval (CI) (−3.4; −0.2), *p* = 0.03) and SUL_peak_ (−1.2, 95% CI (−2.1; −0.3), *p* = 0.01) from scan 1 (prior to treatment) to scan 5. The change in SUV measured on scan 5 and scan 2 or scan 3 was not significantly different. However, the change measured on scan 4 tended to be smaller and more positive (increasing SUV from scan 1 to scan 4) than the changes measured on scan 5 (*p* = 0.04 and 0.03 for SUV_max_ respectively SUL_peak_).

In Table 4 patients are listed according to survival, best response on CT after therapy and changes in early respectively late ADC and SUV. SUV decreases tended to indicate long survival. e.g., all patients with a survival longer than 6 months had an unchanged or decreasing SUL_peak_ and SUV_max_ (not shown) at the early PET/MR scan. On the later scan the majority of patients had a decreasing or unchanged SUV, independently of later response and survival. 

### 3.4. Changes in ADC

Illustration of absolute ADC-values (mean and median) and relative changes for individual patients during the course of therapy are available online (Online Resource 1). Inspection of histograms and Q-Q plots revealed that ADC-data were not normally distributed, thus a log-transformation was applied prior to analyses. Figure 1B illustrates changes in ADC during the study period for individual patients. Table 5 illustrates median values and quartiles for ADC_median_ at each time point for all patients and grouped by first or second cycle of chemotherapy and response. Due to the low number of patients in each group no statistical comparison between groups was performed, but results indicate higher ADC-values in patients scanned during the first treatment cycle as compared to second cycle and in patients with PD compared to patients with SD and PR. There was no significant change in ADC_median_ from scan 1 (prior to treatment) to scan 5 (day 19 and always prior to initiation of the next cycle) (*p* = 0.73). Similarly, the difference in ADC_median_ on scans 2, 3 and 4 as compared to scan 5 was not significant. However, as illustrated in Table 4, a small decrease in ADC was more often seen in patients with a survival of 6 months or less.

## 4. Discussion

This prospective study assessed the feasibility and evolution of FDG-uptake and diffusion in 11 patients with NSCLC scanned up to five times during one cycle of chemotherapy, including a total of 45 PET/MR scans. The study was performed in order to explore the technical feasibility of repeated PET/MR very early during chemotherapy and the existence of multi-parametric patterns, potentially enabling us to predict a response very early during therapy, with a special focus on the possible existence of flare potentially hampering the use of early response evaluation. 

### 4.1. Feasibility 

The study was performed as a prospective feasibility study, generating basic knowledge and methods as a platform for future prospective studies on the use of multi-parametric imaging with PET/MR in tumor response evaluation. The study proves that it is challenging to schedule five scans in three weeks during chemotherapy for patients newly diagnosed with a severe illness and with short life expectancy. Moreover, technical challenges primarily related to the attenuation correction of PET-scan based on MR-Dixon sequence was also experienced, these are reported in detail in a separate technical paper [28]. In short, the Dixon-AC maps were often flawed by artifacts with potential impact on the quantification of FDG-uptake. Thus, Dixon-AC maps should always be inspected prior to any quantification on PET-images. Further, it might be useful to obtain two or more Dixon-AC maps to improve quality of the quantification. Another limitation of the PET attenuation correction performed here is that the flexible MR coils were not considered [31,32]. The coil attenuation depends on axial position, which may vary between examinations, and on distance to the coil. The resulting potential variation in the bias of PET activity values has been estimated to 0.9%–2.2% (standard deviations) corresponding to a random coil positioning for repeated examinations [32]. While this worst-case variation is much smaller than the percent changes observed, coil attenuation correction could be worthwhile for future studies [33].

In 1999 European Organization for Research and Treatment of Cancer (EORTC) published guidelines for measurement of clinical and subclinical tumor response using FDG-PET [34]. In 2009 a new set of guidelines Positron Emission Tomography Response Criteria in Solid Tumors (PERCIST) [29,35] was proposed and even though the acceptance of these criteria have been quite slow, recent publications support the use of PERCIST over the more simple EORTC criteria [36,37,38,39,40,41,42,43,44,45,46,47,48,49]. This study applied, for the first time in a PET/MR setting, PERCIST measurement. Among other things PERCIST requires stable measurements of a reference SUL in the liver. Despite known challenges with attenuation correction [40], we obtained relatively stable reference values in all patients except patient 3, where measurement was hampered due to several liver metastases (Appendix A). With the precaution of this being a very small study, we believe that PERCIST can also be adopted for response evaluation with PET/MR.

### 4.2. Response Patterns and Flare

The rationale for using functional imaging, such as FDG-PET and DW-MRI, is the ability of these modalities to detect functional changes in the tumor tissue prior to any changes observed by anatomical measures [39]. Our findings partly support this hypothesis; all tumors decreased in size during therapy (independent of later response), but these changes were very small and occurred later than observed changes in SUV and ADC. This is also confirmed by another recent study applying FDG-PET/MR for very early response evaluation [41].

It has been reported that SUV values obtained from tumor areas typically decrease following effectively intervention in solid tumors. However, studies have described a temporary increase in SUV, a flare, shortly after chemotherapy administration due to cell swelling, fibroblasts and macrophage infiltration despite therapeutic effect [13,42,43]. Thus, it is currently a widely hold opinion that differentiating between increased FDG uptake due to flare and true disease progression may not be possible early after chemotherapy. The result of this is the current recommendation to postpone an FDG-PET scan to 10–14 days after chemotherapy. In this study we did not find any significant difference between the response measured by SUV or SUL during the first week after therapy (scan 2 and 3) and immediately prior to the next cycle (scan 5, week 3). However, more unstable results were seen on scan 4 (week 2) that could indicate the existence of a flare phenomenon at this time point. This finding indicates that early response evaluation is probably best done immediately after or prior to the next cycle of therapy, and that response measured halfway between two cycles might be unstable. However, as this study is small and exploratory, the significant difference observed at scan 4 could be a chance finding due to multiple testing. Thus, confirmation in larger studies is clearly needed. Response assessment after treatment with the more recently introduced immune check-point inhibitors, was beyond the scope of this paper, nonetheless newer research indicates that early response evaluation may also be possible in this setting despite the risk of flare [44,45].

Earlier studies on DW-MRI reports that ADC values, representing tumor areas, typically increase following effective intervention. However, as reviewed by Galban et al. [46], an initial decrease, followed by an increase in ADC measurements can correlate with a positive response. The initial decrease in ADC values can be a consequence of cell swelling followed by a later cell death causing the ADC values to increase. Later fibrotic changes can again cause ADC values to decrease. Thus, timing of DW-MRI after therapy is crucial, but studies with repeated DW-MRI measurements during chemotherapy are lacking. In this study we did not observe the proposed pattern of initial decrease followed by an increase in ADC. But a decrease in ADC value, both early and late, was observed more frequently in patients surviving 6 months or less. However, as opposed to the majority of previously published studies on early DW-MRI for response evaluation, we did not observe any significant changes in ADC values during the course of treatment. Our results together with the results of similar study done in patients with lymphoma, indicates that response is probably seen slightly later with DW-MRI as compared to FDG-PET [42]. Early response evaluation in the literature tends to be done after, rather than during the first cycle of chemotherapy [24,47,48,49] and together with the relatively low number of observations in our study, this could be a possible explanation for this difference. 

The changes observed at the very early as well as later scans in this study with regard to both SUV and ADC, were generally small and smaller than the recommended cut-off points of respectively 30% in PERCIST and the applied 25% for ADC [29,30]. Under due consideration of the test-retest stability of the individual methods, this indicates that different cut-off points should be tested for very early response evaluation, which has also been suggested in the original EORTC response criteria and most recently by Cho and colleagues [34,45]. 

In this study patients were included prior to the first or the second cycle of chemotherapy. Our results indicate that, for both FDG-PET and DW-MRI, the response signal measured during the first cycle of chemotherapy is more pronounced than during the second cycle. This is in line with the findings reported by Nahmias and colleagues. Performing up to seven FDG-PET scans during the first two cycles of chemotherapy, they found that responses are achieved after one cycle of chemotherapy and that responses are less predictable with further chemotherapy cycles [50].

In summary, even though this study included a relatively small number of patients and results should be interpreted with caution, our results raise several important points: Repeated multi-parametric imaging by PET/MRI during chemotherapy in newly diagnosed lung cancer patients is challenging. Quantification of FDG-uptake on PET/MR requires caution with regard to the quality of attenuation correction; nonetheless the application of PERCIST appears feasible. Response evaluation shortly after initiation of therapy is concordant with later evaluation and probably more reliable than evaluation midway between chemotherapy cycles. Last, but not least, responses are likely to be achieved and more readily measured during or after the first cycle of chemotherapy rather than during subsequent cycles. We are currently initiating a larger study using PET/CT to confirm or disprove these findings in patients treated with chemotherapy and also with other treatment regimens, i.e., immune checkpoint inhibitors. 

## Figures and Tables

**Figure 1 diagnostics-09-00035-f001:**
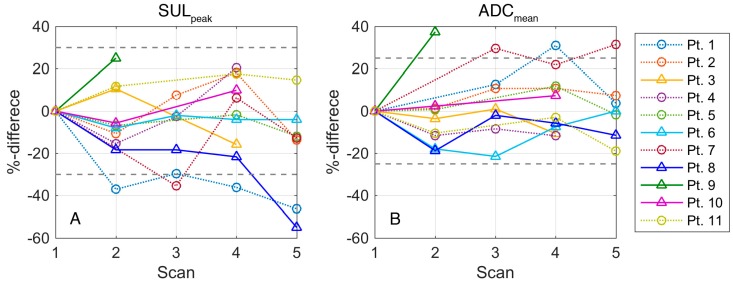
Changes in (**A**) FDG-uptake measured as SUL_peak_ and (**B**) ADC_median_ during the study period. Changes are presented for individual patients as percent difference with scan 1 as baseline. Dotted lines with circular markers indicate that the patient obtained stable disease or partial response (patient 1). Solid lines with triangular markers indicate patients with progressive disease.

**Table 1 diagnostics-09-00035-t001:** Patient characteristics.

Patient Data	N (%)
Sex (male/female)	7 (64)/4 (36)
Mean age (range)	62 years (52–73)
Histology (Adenocarcinoma/squamous cell carcinoma)	7 (64)/4 (36)
Chemotherapy cycle (first/second)	6 (54)/5 (46)
Response at CT evaluation (CR /PR /SD /PD) *	0 (0)/1 (9)/5 (45)/5 (45)

* Response according to RECIST 1.1: CR, Complete Response. PR, Partial Response. SD, Stable Disease. PD, Progressive Disease.

**Table 2 diagnostics-09-00035-t002:** Timing and completion of chemotherapy and PET/MR scans.

Day	−3	−2	−1	0	1	2	3	4	5	6	7	8	9	10	11	12	13	14	15	16	17	18	19	20	21
week					1							2							3						
Car ^a^				**X**																					
Vin ^b^				**X**								**X**													
PET/MR	Scan 1				Scan 2					Scan 3		Scan 4								Scan 5					
N (%)	11 (100)				10 (91)					7 (64)		10 (91)								7 (64)					
Med ^c^	−1 (−3 to 0)				1 (1 to 3)					6 (6 to 7)		10 (8–11)								19 (16–21)					

^a^ Treatment with carboplatin intravenous (i.v). ^b^ Treatment with oral vinorelbine (p.o). ^c^ Median day and range for performance of PET/MR.

**Table 3 diagnostics-09-00035-t003:** FDG-uptake in tumor during chemotherapy.

**SUV_max_** **Mean (SD)**	**All Patients** ***n* = 11**	**1. Cycle** ***n* = 6**	**2. Cycle** ***n* = 5**	**PR ^a^** ***n* = 1**	**SD ^b^** ***n* = 5**	**PD ^c^** ***n* = 5**
**Scan 1**	9.9 (3.6)	10.6 (2.9)	9.0 (4.6)	15.2	9.8 (4.4)	8.8 (2.2)
**Scan 2**	8.5 (1.9)	9.8 (1.2)	6.5 (0.9)	10.6	7.6 (2.2)	8.7 (1.8)
**Scan 3**	8.7 (2.1)	10.4 (1.7)	7.4 (1.4)	10.9	9.1 (2.9)	7.6 (0.9)
**Scan 4**	9.9 (3.4)	10.8 (1.9)	9.0 (3.4)	10.3	10.9 (3.8)	8.5 (3.2)
**Scan 5**	8.6 (2.8)	9.0 (1.3)	8.2 (4.5)	8.3	9.9 (2.6)	6.3 (3.0)
**SUL_peak_** **Mean (SD)**	**All Patients** ***n* = 11**	**1. Cycle** ***n* = 6**	**2. Cycle** ***n* = 5**	**PR** ***n* = 1**	**SD** ***n* = 5**	**PD** ***n*= 5**
**Scan 1**	5.8 (2.3)	6.5 (2.4)	5.1 (2.0)	10.8	5.5 (1.9)	5.1 (1.5)
**Scan 2**	5.0 (1.2)	5.7 (1.0)	4.1 (0.7)	6.8	4.6 (1.3)	5.0 (1.0)
**Scan 3**	5.3 (1.5)	6.5 (1.4)	4.4 (0.7)	7.6	5.3 (1.7)	4.5 (0.7)
**Scan 4**	5.8 (1.8)	6.6 (1.3)	5.0 (2.0)	6.9	6.1 (1.9)	5.1 (1.9)
**Scan 5**	5.0 (1.4)	5.4 (0.5)	4.5 (2.2)	5.8	5.4 (1.2)	3.8 (1.5)

^a^ Partial Response according to RECIST 1.1, measured on CT after two-three cycles of chemotherapy. ^b^ Stable Disease according to RECIST 1.1, measured on CT after two-three cycles of chemotherapy. ^c^ Progressive Disease according to RECIST 1.1, measured on CT after two-three cycles of chemotherapy.

**Table 4 diagnostics-09-00035-t004:** Survival and early response on PET/MR.

Patient	Survival *	RECIST ^	Early ADC ^†^	Early SUV ^‡^	Late ADC ^§^	Late SUV **
**5**	Alive	NC				
**4**	15	NC				
**1**	12	PR				
**10**	20	PD				
**2**	10	NC				
**7**	8	NC				
**11**	6	NC				
**8**	3	PD				
**9**	3	PD			NA	NA
**6**	2	PD				
**3**	1	PD				

* Months from inclusion (date of chemotherapy) to death. ^†^ ADC_median_ at scan 2 or 3 after chemotherapy. Red = >25% decrease, Green = >25% increase, beige = no change. Arrows indicates changes smaller than 25% but larger than 10%. Please note increasing ADC indicates response. ^‡^ SUL_peak_ at scan 2 or 3 after chemotherapy. In patient no. 9 SUL_peak_ and SUV_max_ increased > 25%, Red = >30% increase, Green = >30% decrease, beige = no change. Arrows indicates changes smaller than 30% but larger than 10%. decreasing SUV indicates response. ^§^ ADC_median_ at scan 4 or 5 compared to baseline. Red = >25% decrease, Green = >25% increase, beige = no change. Arrows indicates changes smaller than 25% but larger than 10%. ** SUL_peak_ at scan 4 or 5 compared to baseline. In patient no. 9 SUL_peak_ and SUV_max_ increased > 25%, Red = >30% increase, Green = >30% decrease, beige = no change. Arrows indicates changes smaller than 30% but larger than 10%. ^ PD = Progressive Disease; CR = Complete response; change; NA = Not applicable.

**Table 5 diagnostics-09-00035-t005:** Diffusion in tumor during chemotherapy.

**ADCmedian ^††^** **Median (Q_1_; Q_3_)**	**All Patients** ***n* = 11**	**1. Cycle** ***n* = 6**	**2. Cycle** ***n* = 5**
**Scan 1**	1153 (1081; 1207)	1190 (1038; 1348)	1122 (1029; 1454)
**Scan 2**	1201 (964; 1439)	1216 (1048; 1547)	1058 (906; 1339)
**Scan 3**	1312 (1133; 1401)	1312 (1231; na **)	1267 (954; 1640)
**Scan 4**	1315 (987; 1483)	1350 (1126; 1535)	1119 (933; 1486)
**Scan 5**	1273 (1134; 1553)	1230 (1147; 1647)	1421 (934; na)
**ADCmedian** **Median (Q_1_; Q_3_)**	**PR** ***n* = 1**	**SD** ***n* = 5**	**PD** ***n* = 5**
**Scan 1**	1093	1153 (1029; 1196)	1194 (998; 1762)
**Scan 2**	- *	1117 (906; 1212)	1425 (989; 1547)
**Scan 3**	1231	1312 (894; na)	1390 (1133; na)
**Scan 4**	1431	1313 (991; 1334)	1321 (954; 1650)
**Scan 5**	1134	1230 (997; 1385)	1662 (1553; na)

^††^ Unit 10^−6^ mm^2^/s.* No data available due to technical challenges.** Not applicable.

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
