# Peer review of "Very Early Response Evaluation by PET/MR in Patients with Lung Cancer—Timing and Feasibility"

_diagnostics, 2019, doi:10.3390/diagnostics9010035_

Reviewer 1 Report

In this manuscript the authors present preliminary data on FDG PET/MT imaging as a very early treatment response assessment (1, 2 and 3 weeks post treatment start) during chemotherapy in lung cancer patients. This is the first study that conclusively tries to answer the question on how early FDG imaging could make sense as treatment follow up and should be interesting for the readers of Diagnostics. The study data is presented as is without whitewashing and in full awareness that patient numbers are too small to state any significance of the data. The authors however conclude that a larger study is feasible, and I’m certainly curious to see how that is going to work out. This manuscript can be published as is with minor spell checking (eg Line 169-170 ‘ADCmean correlates with ADCmean’.

Author Response

Dear Sir,

Thank you for considering our manuscript for publication in Diagnostics. We value the feedback from the reviewers and have changed the manuscript accordingly. Please refer to our point-by-point response below.

Thank you for the positive comment. We have changed the above as required and performed further spell checking.

Reviewer 2 Report

The manuscript addresses the topic of early therapy response monitoring for cancer patients. The fact that combined PET/MR is used for lung cancer makes the approach challenging but also allows for potentially new findings that could point towards a further application of this new dual modality imaging technology. Before publishing, a few methodology aspects should be clarified.

Major (MA) 1. In line 100, the authors state that flexible MR coils are used and in line 105, it is mentioned, that Dixon sequences are used for the generation of the attenuation maps for PET. It is not clear from this description, if the described method for attenuation correction includes the attenuation correction for the coil also, if the attenuation correction for the coil is done with another method, or if attenuation correction for the coils is just omitted. Position of the coil may vary from scan to scan (also not described) and depending on the relative position of the coil with respect to the ROI (i.e tumor), attenuation by the coil may have a large impact on the quantification of the tracer uptake. Please explain, how the attenuation and its potential impact on the results was addressed.

MA 2. In lines 169 it is stated that only SUVmax and SULpeak are used, and SUVmean is omitted since it is highly correlated with the other two measures. However, it was not stated, why the choices was to use SUVmax and SULpeak instead of SUVmean. Please, give a motivation for this. Further, it should be explained, what SUVmax and SULpeak exactly means, and how it is extracted. Since PET images have much higher statistical noise, sometimes SUVmax is also an average over several voxels to lower the impact of noise on the value. In this context, it should also be mentioned, if the PET image was smoothed whit a gaussian filter, if scatter correction was applied, what was the image voxel size and which image reconstruction method and settings were used. 

MA 3. In lines 204, it is explained, that normality of ADC data was assessed by inspection. Easy statistical tests exist and it is not clear, why they cannot be applied here. Further, the author state that, if the inspection revealed non-normality, the logarithm is applied to the data. What is the intention and background of this method? The description suggests, that the authors expect the ADC data in these cases to be log-normal distributed, and to transform then into a normal distribution. This approach would introduce unnecessary variance and also makes ADC data with applied log and without applied log incomparable. Possible tendencies and dependencies could be masked. A clear and consistent approach should be used here. 

MA 4. Statement in lines 210ff is not clear. Author state, that changing ADC values when comparing scans of first therapy cycle to second cycle. But they also state, that there was no significant change when comparing Scan 1,2,3,4 to scan 5. Since it is not clearly stated if a chemo cycle lasts exactly 1 week (although it could be deduced from table 2), scan 5 could also be cycle 2. Furthermore, in line 251, it is stated that these result support the hypothesis that DW-MRI is able to detect early functional changes, which is not well supported by the ADC results, especially bearing in mind MA 3. 

Minor (MI) 1. Unclear wording in line 50: "and who should be spared the side effects"

MI 2. NC and NA (not applicable? ) not explained in table 4.

MI 3. Dangling footnote (line 247) .

MI 4. Line 283: response i probability

Author Response

Dear Sir,

Thank you for considering our manuscript for publication in Diagnostics. We value the feedback from the reviewers and have changed the manuscript accordingly. Please refer to our point-by-point response below.

(MA) 1. Thank you for highlighting this potentially relevant issue. PET attenuation correction was performed without considering the flexible MR coils. Coil attenuation can to potentially bias PET activity values with up to 20% directly below the coil (Paulus et al.). As the reviewer rightly points out, this bias depends on axial position along the coil, and the latter might vary between examinations. The variation in the bias of PET activity values due to axial coil position for the standard attenuation correction employed also in our study, was in the detailed phantom study (Paulus et al.) found to be 0.9% - 2.2% (standard deviations), depending on transaxial position. This variation was however partly due to slow variations of the bias over the entire axial distance of 19.6 cm considered. We estimate that such a variability in coil positioning between exams is unrealistic. Still, in a worst-case scenario of lesion position in the most sensitive area and coil positioning completely random with respect to lesion, this effect would add 2% variability (standard variation) to the time series; much smaller than the changes observed. A mention of this has been added to the Discussion.

MA 2. In order to clarify the issues raised above the following details were added to the manuscript:

“PET data was reconstructed using 3-dimensional ordinary Poisson ordered-subset expectation maximization with 3 iterations, 21 subsets, and 4-mm gaussian postfiltering on 344 · 344 · 224 matrices with a voxel size of 2.1 · 2.1 · 2.0 mm.”

The extraction and meaning of SUVmax and SUVpeak is described in the method-section. We have kept this brief in order to keep within the word-limit, but we have now added the formula for completion:

For calculating SUVmax,r is the maximum radioactivity activity concentration [kBq/ml] measured by the PET scanner in any voxel within the tumor.  For SUVmean r is the mean radioactivity activity concentration [kBq/ml] measured in the tumor. a′ is the decay-corrected amount of injected radiolabeled FDG  [kBq], and w is the weight of the patient in grams.

For SULpeak, r is the radioactivity concentration averaged within a 10-mm-diameter spheric ROI positioned by the MIRADA software within the tumor so as to maximize the enclosed average. Instead of normalizing to the weight of the patient the lean body mass (LBM) of the patient was inserted in the above formula. 

MA 3. Unfortunately, simple tests of normality such as the Shapiro-Wilk or Kolmogorov-Smirnov test require that observations are independent and thus cannot be applied to repeated measurements data. For this reason, normality was assessed from residual plots (studentized and scaled residuals in the output from sas’ proc mixed). As appeared ADC data was skewed, and log-transformation substantially improved the fit to the multivariate normal distribution. We chose to make the transformation to reduce the influence of outliers even though the skewness might be a spurious finding (which is likely considering the small sample size). While we agree with the reviewer’s concerns, on the other hand not transforming would render analyses invalid if the distribution truly is log-normal (in particular as the mixed model implicitly imputes missing data from the normal model), so there is no fool proof solution here. However, since this is a feasibility study, it is clear that results should be interpreted with caution and larger studies are needed to make firm conclusion including on whether or not data is normally or log-normally distributed.

MA 4. Sorry if we did not state this clearly. Scan 5 was always performed prior to the next cycle. We have rephrased for further clarification and modified our statement (previous line 210 respectively 251, now 243)

(MI) 1. Wording changed for clarification.

MI 2. Thank you, we have now explained the above in the text beneath the table.

MI 3. The problem has been solved.

MI 4. The misspelling has been corrected.

Reviewer 3 Report

1. In this study, the authors performed 5-time scanning of PET/MR in patients undergoing chemotherapy for lung cancer. Even though their efforts are valuable, I am not able to understand what the authors intended to demonstrate. Do authors want to tell that 5-time scanning is needed for precise evaluation? Do authors want to optimize the timing of PET/MR exam in order to reduce the number of scanning? Please clarify the point and re-construct the manuscript.

2. The sentence in Abstract Results: "The changes in FDG-uptake measured within the first week of therapy were not different from changes measured after 3 weeks, but changes measured at week 2, differed from the change measured at week 3." is too complicated to transfer the message. Please rephrase to transfer the authors' message more straightly.

3. In Introduction, "Compared to PET/CT the PET/MRI system enables radiation dose reduction, improved soft-tissue contrast and most importantly simultaneously acquisition of information on tumor anatomy and several functional parameters [16, 18, 19]. " Although the authors emphasized simultaneous scanning of PET and MRI using the words of 'most importantly', I did not find that the authors took advantages of simultaneous scanning. Instead, authors analyzed PET and MRI separately, which can be achieved with standalone PET and standalone MRI scanners.

4. Figure1 shows solid and dotted lines, but it is difficult to distinguish them in the current image quality. Please consider using other signs.

5. In Discussion, in 4.1 Feasibility subsection, "The study proves that it is possible, but challenging,.." I do not feel 5-time scanning is clinically accepted unless there are significant clinical impacts. I wonder if the authors found it was challenging as the result of the current study (if so, the results should be shown in Results section), or the authors had already known it was challenging.

Author Response

Dear Sir,

Thank you for considering our manuscript for publication in Diagnostics. We value the feedback from the reviewers and have changed the manuscript accordingly. Please refer to our point-by-point response below.

1. Thank you for this valuable comment. We are very sorry if the aim doesn’t come thru and have revised the introduction and discussion accordingly.

2. The sentence has now been rephrased: “There was no difference in the FDG-uptake measured 1 or 3 weeks after therapy, but uptake measured 2 weeks after therapy differed from measurements at week 3”

3. The above sentence has been rephrased.

4.Thank you, we have updated the figure. 

5. 5-time scanning – being on PET/MR or PET/CT – is unlikely to ever be clinical routine. However, studies like this is necessary for the community to get a better understanding of the dynamic of FDG-uptake during therapy. This knowledge (of which the current study is a small but significant step) is crucial to improve our current practice with regard to response evaluation in solid tumors.

We did indeed expect the study to be challenging with regard to clinical feasibility, but we were surprised by the magnitude of the technical challenges associated with repeated scanning on PET/MR. As mentioned in the manuscript this have been presented and discussed in details in a technical paper (Olin et al, J Nucl Med, 2018).

Reviewer 4 Report

This paper assesses the feasibility and evolution of FDG-uptake and diffusion in 11 patient.

The study is well preformed and the manuscript nicely written. The only major downside is the small number of patients on basis of whom the conclusions are based. However, I do agree such a complex protocol including up to 5 PET/MRI scans for each patient is quite difficult to perform in practice. 

Minor issues:

1. P2,L60: an overview is given of problems during FDG imaging, the authors should include statements explaining the facts that: FDG requires standardization of the scanning protocol SUV can strongly be by various factors (such as fasting period, time-interval, calibration, biological factors such as differences in tumors metabolism, endogenous ligand, treatment affecting metabolism etc). Include references.

2. P2, L71-72: PET/CT is compared to PET/MRI, nothing about attenuation problems is mentioned here, which is a generally know problem on PET/MRI systems. Include some references.

3. P3, L106-107: need to be more specific why simply repeating scans twice would resolve artifact. Are not the artifacts partly patient and partly scanner related?   

4. P5, L187 & P6, L213 & P3, L128: The estimated minimum requirement was clearly not enough to for statistical comparison. This needs to be discussed what went wrong in the initial estimate. 

Author Response

Dear Sir,

Thank you for considering our manuscript for publication in Diagnostics. We value the feedback from the reviewers and have changed the manuscript accordingly. Please refer to our point-by-point response below.

1. We absolutely agree and have added a sentence and some references on this issue. However, the word limit does not allow us to go into details.

2. We have now mentioned this in the introduction and added a reference.

3. Repeating scans twice will not eliminate artifacts. However, in the study by [Olin et al.] SUV variation, caused by differences MR-AC maps, resulted mainly from respiratory motion. This is patient related and by repeating a scan we increase the likelihood of obtaining at least one MR-AC map without or less affected by this artifact. We have specified that artifacts such as respiratory motion is the basis for choosing the MR-AC map for attenuation correction.

4. This study is, to the best of our knowledge, the first with repeated PET/MR scans very early in therapy and as such designed as a feasibility study. We planned to use a linear mixed model for data analysis and for this purpose 10 patients were considered an absolute minimum. We agree, that more patients would have been a strength but one the other hand we have, as planned, used the knowledge provided by this study to plan a larger study (with fewer time points and more patients). This study is currently on-going. 

Round  2

Reviewer 2 Report

All issues from the first review round have been sufficiently addressed. 

Reviewer 3 Report

The authors modified the manuscript, but in their Response 5, 'However, studies like this is necessary for the community to get a better understanding of the dynamic of FDG-uptake during therapy.' but as they admitted, the number of the patients was small. I feel it is too small to reach the conclusion 'This pilot study indicates that response evaluation shortly after initiation of chemotherapy appears concordant with later evaluation and probably more reliable than evaluation midway between cycles.' Also, Figure 1 is still not very clear how to interpret the data. Authors should consider different presentation.